# SARS-CoV-2 Vaccination and Anaphylaxis: Recommendations of the French Allergy Community and the Montpellier World Health Organization Collaborating Center

**DOI:** 10.3390/vaccines9060560

**Published:** 2021-05-27

**Authors:** Luciana Kase Tanno, Frédéric Berard, Etienne Beaudoin, Alain Didier, Pascal Demoly

**Affiliations:** 1Division of Allergy, Département de Pneumologie et Addictologie, University Hospital of Montpellier, 34295 Montpellier, France; pascal.demoly@inserm.fr; 2WHO Collaborating Center for Classification Scientific Support, University Hospital of Montpellier, 34295 Montpellier, France; 3Desbrest Institute of Epidemiology and Public Health, UMR UA-11, INSERM University of Montpellier, 34093 Montpellier, France; 4Hospices Civils de Lyon, Universite Claude Bernard lyon I, Inserm U1111-CIRI, 69495 Lyon, France; frederic.berard@chu-lyon.fr; 5Regional Institute for Allergic and Environmental Diseases-Clinical Immunology, Metz Regional Hospital, 57000 Metz, France; beaudouin.etienne@wanadoo.fr; 6Pôle des Voies Respiratoires, Hôpital Larrey, CHU de Toulouse, CEDEX 9, 31059 Toulouse, France; didier.a@chu-toulouse.fr; 7Centre de Physiopathologie Toulouse Purpan, INSERM U1043, CNRS UMR 5282, Université Toulouse III, CEDEX 3, 31024 Toulouse, France

**Keywords:** allergy, anaphylaxis, COVID-19, recommendations, SARS-CoV-2 vaccination, vaccine

## Abstract

Vaccines against COVID-19 (and its emerging variants) are an essential global intervention to control the current pandemic situation. Anaphylactic reactions have been reported after SARS-CoV2 RNA vaccines. Anaphylaxis is defined as a severe life-threatening generalized or systemic hypersensitivity reaction. This risk is estimated at 1/1,000,000 in the context of vaccine safety surveillance programs. The COVID-19 vaccination is rolling-out vastly in different courtiers and surveillance programs are key to monitor severe adverse reactions, such as anaphylaxis. Anaphylaxis due to vaccine is extremely rare and specific cases should receive individualized investigation and care. The here presented recommendations and follow-up from the French allergy community and the Montpellier WHO Collaborating Center in order to support the vaccination program and intends to support to healthcare professionals in their daily basis.

## 1. The Scope of the Problem

### 1.1. The SARS-CoV-2 Vaccination: Essential to Control the COVID-19 Pandemic

As of 13 April 2021, a total of 135,057,587 cases of coronavirus disease 2019 (COVID-19) and 2,919,932 associated deaths were reported globally by the World Health Organization (WHO) [1]. Long-term sequelae manifestations and serious complications were reported among COVID-19 survivors, including individuals who initially presented with mild acute illness.

Vaccines against COVID-19 (and its emerging variants) are an essential global intervention to control the current pandemic situation. COVID-19 vaccination programs have been launched in many countries in early 2021. On 11 December 2020, the USA Food and Drug Administration (FDA) issued an Emergency Use Authorization (EUA) for Pfizer-BioNTech COVID-19 vaccine (Pfizer-BioNTech vaccine) [2,3]. This followed by the authorization of Moderna mRNA (ribonucleic acid)-1273 vaccine (Moderna vaccine) for use by other regulatory agencies, such as the European Medicine Agency (EMA), UK Medicines and Healthcare Products Regulatory Agency (MHRA), Israel Ministry of Health among others [3,4,5,6]. As of 23 December 2020, first doses of Pfizer-BioNTech vaccine had been administered in the USA. Until April 2021 reports of 4393 (0.2%) adverse events to the Pfizer-BioNTech vaccine have been reported, and among these, 175 cases of severe allergic reactions, including anaphylaxis [2].

In the UK, there was an initial concern that patients with atopic diseases might be more likely to develop allergic reactions to COVID-19 vaccines. The UK issued an advisory statement which listed prior anaphylaxis to a vaccine, medication, or food as a contra-indication [7]. This recommendation was not followed by USA FDA nor the US Centers for Disease Control and Prevention (CDC), although there is a recommendation of 15 min observation following the vaccination [2]. These recommendations generated a fear of receiving the mRNA COVID-19 vaccine in the population and was the reason for many inquiries from health professionals dealing with COVID-19 and vaccination programs.

Later, on 30 December 2020, the UK Minister of Health reviewed the previous recommendations, limiting the contra-indications to patients who reacted to the first injection or who had a confirmed allergy to any component of the vaccine. In France, the HAS (High Authority of Health) reviewed the recommendation to the m-RNA Pfizer-BioNTech vaccine and published: “Vaccination is contraindicated only in the case of history of immediate allergy to one of the components of the vaccine, or in case of history of serious immediate reaction occurring within 6 h after a first injection (manifestation suggesting severe anaphylactic reaction with respiratory, skin, digestive or hemodynamic manifestations); the occurrence of a moderate skin manifestation is not a contraindication. History of allergy to other substances, such as hymenoptera venom, inhalant allergens, foods, regardless of their severity, are not a contraindication to COVID-19 vaccine”.

### 1.2. The Allergy Epidemic

With more than a quarter of the general population suffering from allergies and soon half according to the WHO, the food and pharmaceutical industries in general and vaccines in particular are increasingly faced with the risk of allergic manifestations to their products, not detected during the development phase, sometimes documented and confirmed but far from it. They also have to deal with false reactions, fears that may or may not be justified, but the impact of which can be considerable medically and economically. This increase in the frequency of allergic diseases is real and concerns all clinical forms, in particular the most serious, the one that is life-threatening: anaphylaxis. Therefore, this is the context of vaccine-induced anaphylaxis, especially in the case of COVID-19 vaccines. The introduction of a new vaccination is always the subject of debate, even inducing reactions of rejection. This problem echoes the significant public mistrust of the vaccine program for diseases deemed benign, mistrust amplified by the Internet, false rumors, and deleterious media frenzy. More than ever the science of real data must prevail and communication must be mastered and scientific.

Anaphylaxis is defined as a rapid onset, life-threatening, systemic hypersensitivity reaction by compromising the cardiovascular and respiratory systems. The mast cell activation that underlies its pathophysiology often results from immunological mechanisms (IgE-dependent) but not always. There are cases of non-immune-mediated anaphylaxis in which manifestations can be developed with no IgE-mediated mechanism, such as in the context of mast cell activation syndrome. This multifaceted disease can occur at any age and in varying degrees of severity [8]. European data indicated incidence rates of all-cause anaphylaxis ranging from 0.3 to 7.9/10^5^ people/year, with an estimate that 0.3% (95% CI 0.1–0.5) of the population will suffer from anaphylaxis at some point in their life, known as the “lifetime anaphylactic risk” [8,9]. The calculation made at the Montpellier University Hospital Center is 0.32/10^5^ (95% CI 0.28–0.65) [10]. Although it is a cause of death well known by physicians, anaphylaxis has never been properly monitored due to the difficulties of classification and coding in the different versions of the WHO International Classification of Diseases (ICD). For this reason, anaphylaxis has never been considered underlying cause of death in death certificates leading to under notification, until recently due to the construction and implementation of its 11th version, ICD-11 by the WHO Collaborating Center in Montpellier with the WHO governance for international classifications [11,12]. Strikingly however, it has been listed for a long time in vaccine safety surveillance programs [13].

## 2. Current Status: What Do We Know?

### 2.1. Impact of Vaccine Safety in Immunization Programs

The WHO has published on its website [1] a 5-phase introduction to vaccine safety based on Chen et al. [13]. In the phase before the introduction of vaccines, morbidity and mortality from infectious diseases that are preventable today are high. Since vaccines did not exist, there were no adverse events. After the introduction of an effective vaccine, the incidence of new cases decreases, but at the same time adverse effects, real or perceived, appear and can become problematic. It is when the benefits of the vaccine are most evident and immunization coverage is highest that concerns about vaccine safety coming from sporadic rare adverse events are more likely to increase among the general public.

This increased attention to adverse effects around only a few cases was intensified by the media and led to a loss of public confidence in the vaccine, then a drop in vaccination coverage and finally a reoccurrence of the disease sometimes at epidemic levels, as was seen with measles. In the face of a re-emergence of disease or the availability of a new vaccine, the public regains confidence and accepts the vaccination again, resulting in again high vaccine coverage and reduction of disease to previous low levels. This is the case with whooping cough, for example. For diseases eradicated by vaccination, the use of vaccines can be discontinued, thus eliminating the risk of an adverse event. This was the case with smallpox.

### 2.2. Vaccine Safety Surveillance Programs

To ensure the safety of vaccines, side effects need to be detected early, evaluated and acted on in a way that maintains public confidence. The passive surveillance system supported by the CDC and the FDA in the United States is the Vaccine Adverse Event Reporting System (VAERS). The term “severe” includes hospitalization or prolongation of hospitalization, persistent or significant disability, or endangerment of life. In France, the side effects of drugs are reported by health professionals to the National Agency for the Safety of Medicines and Health Products (ANSM) via the Regional PharmacoVigilance Centers [14]. Pharmaceutical industries also have this obligation. ANSM reports cases to EMA, the European Medicines Agency (EudraVigilance), and to WHO (VigiBase) are more robust international pharmacovigilance databases. These programs provide incidence data, they consider the temporal association with be causal, yet the contrary examples in drug [15] and vaccine [16] allergies are numerous. Current in France (April 2021), more than 13,610,000 COVID-19 vaccine injections (among Pfizer-BioNTech, Moderna and AstraZeneca) were administrated, with few reports of severe adverse effects [17]. Early safety monitoring of the Pfizer-BioNTech vaccine detected 21 cases of anaphylaxis after reported administration of 1,893,360 first doses of Pfizer-BioNTech vaccine (11.1 cases per million vaccine doses administered) as well as cases of less severe non-anaphylactic reactions, based on USA data on 14–23 December 2020 [2].

Population-based studies and surveillance national programs do not count as having the specific investigation to confirm (or not) allergies and understand the mechanisms involved. However, clinical complaints that arise immediately after the administration of a vaccine, whether or not compatible with an allergic reaction, have a significant impact on the public’s perception of vaccines and their willingness to be vaccinated more.

### 2.3. Anaphylaxis

Anaphylaxis is clinically known as a systemic hypersensitivity reaction characterized by rapid onset and the potential to endanger life through airway, breathing or circulatory involvement. It is usually, although not always, associated with skin and mucosal changes [8]. Its heterogeneous clinical presentation and sudden occurrence in virtually any setting without warning hampers the prompt recognition and treatment of this condition, increasing the risk of death.

Although severe, anaphylaxis due to a vaccine is extremely rare, estimated at 1/1,000,000, and specific cases should receive individualized investigation and care [16,17,18].

### 2.4. Immediate Allergic Side Effects of Vaccines

Immediate allergic manifestations (up to 4 h after injection) to vaccines are extremely rare, estimated between 1/50,000 and 1/1,000,000 [18]. Symptoms and signs are most often cutaneous, generally limited to the site of application and relatively minor (erythema, pruritus) but sometimes multi-systemic (anaphylactic). Post-vaccine anaphylaxis, the most worrying clinical form, is estimated at 1/1,000,000. These are estimates because most of them have not been explored, nor accurately described, or even reported. The differential diagnoses are numerous, with mainly vagal discomfort, panic reactions following normal or exaggerated reactions at the injection site (in subjects already immune). They must be eliminated. These reactions can in theory be IgE-dependent due to a component of the vaccine (vaccine antigens, residual contaminants used for the culture of the infectious agent, additives or excipients) or non-allergic by non-specific histamine release [16,18,19].

The frequency of anaphylaxis [18] varies from vaccine to vaccine as follows: DPT (diphtheria, pertussis and tetanus) (0.36/100,000, usually due to the vaccine agent), influenza (0.08/100,000, exceptionally due to ovalbumin), MMR (measles, mumps and rubeola) (0.18/100,000, less and less due to the porcine gelatin stabilizing it, never to the egg proteins it actually barely contains). Anecdotal cases related to certain other vaccine components have been described [18,19] with the products of Saccharomyces cerevisiae, the latex (no longer), the sugar galactose-α 1,3 galactose (alpha-gal) contained in the gelatin, chicken proteins (for vaccines grown on chicken fibroblasts). No preservatives, adjuvants, or unfiltered antibiotics from cell cultures have been directly and documented in these anaphylactic cases.

Polyethylene glycols (PEG) or macrogols (and their derivatives such as polysorbates) have recently entered vaccines, for example the COVID-19 vaccine BioNTech BNT162b2. They are present in the textile industry, paper, leather, food, certain cosmetics as thickeners or solvents [20]. They are also present in many drugs where they stabilize the dosage form, increase water solubility, skin penetration, or prolong the plasma half-life. They are also used as an active ingredient (an osmotic laxative). They are hydrophilic polymers with CAS number 25322-68-3 produced by polymerization of ethylene oxide (H (OCH2CH2) nOH), of variable molecular weights (from 200 to 35,000 g/mol) depending on the number of ethylene oxide units present. The higher the molecular weight, the less digestive absorption is possible (complete for those with low molecular weight <400 g/mol to less than 10% for those >3300 g/mol). Likewise, only those <3300 g/mol are absorbed through healthy skin. They are considered to be biologically inert, and no documentation exists for example concerning a possible non-specific activation of blood basophils or tissue mast cells; in other words, they are not histamine liberators. On the other hand, they can induce immunological reactions and cross-react with polysorbates [20,21]. A systematic review of the literature found 74 reactions in 37 patients, documented between 1977 and 2016 [20], all adults, three quarters anaphylactic, 8 times out of 10 after oral absorption, half due to PEG laxative but never after vaccination in this series, since Pfizer’s BNT162b2 vaccine is the first to contain it. The six cases after injections (intra-articular, intramuscular, intravenous) were all anaphylactic. It would appear from these cases that molecular weight really matters but that no triggering concentration emerges and that each patient responds to varying doses and concentrations. In a more recent publication, Stone et al. [22] questioned the basis of the drug side effects of the FDA from 1989 to 2017 and found 53 cases of anaphylaxis with PEG 3350 used as a laxative. They also explored two of their patients who both reacted during a preparation for colonoscopy and after infiltration of methylprednisolone acetate all containing PEG > 3000, as well as six controls exposed and showed the interest of the prick test to different PEG as well as the possibility of cross-sensitization with polysorbate 80. Anti-PEG IgG and IgE were found in these two patients (and not in the controls) and subsequently in six other patients who presented anaphylaxis when using a drug containing PEG for the first time, with the aim of developing a commercial biological test [21]. IgG was present in 5–9% of over 1700 control sera tested, IgM in 6% of almost 1000 sera tested and IgE in less than 1‰ of over 2000 sera tested. The majority of cases of PEG-induced anaphylaxis are more severe, requiring more than one dose of epinephrine. Reports of anaphylaxis during PEG skin test in patient with personal history of vaccine-induced anaphylaxis (suspected PEG) should be considered for allergy work-up.

Monomethoxypolyethylene glycol (MPG) has been used for the formulation of several drugs, e.g., interferons, without any major general reactions reported. In the 1980-ties, Pharmacia Diagnostics developed a series of mPEG modified allergen extracts. The products were withdrawn due to the development of positive skin reactions as reported by Freddie Hargreaves group at McMasters [23] and Mosbech in Copenhagen [24] and the non-finalized ongoing trials summarized by Dreborg and Åkerbom [25]. No severe reactions occurred. However, IgG anti-PEG IgG as well as in some cases even IgE antibodies developed.

The need for investigation in order to affirm or deny and for scientific control of information is great, at the risk of seeing false rumors arising and compromising a vaccine program. The two recent cases occurring in England on the second day of the start of the vaccination program in this country of an anaphylactic reaction following the first injection of an RNA vaccine directed against SARS-CoV2 are educational. The two health professionals targeted had a history of severe allergy (excluding vaccine). The English tabloids seized on it and the announcement of the public authorities was not able to calm the spirits, contrary to the immediate statement of the French Society of Allergology (Figure 1). Indeed, there is no scientific support for banning a vaccine for anyone who has suffered from anaphylaxis, whatever the cause in their life, unlike true allergies to one of the compounds in a vaccine [16,18], as the French Haute Autorité de Santé reminded in 2018 [26]: “A history of allergy, even serious, is not in itself a vaccine contraindication unless the offending product is a vaccine. Likewise, a severe allergy to a vaccine does not contraindicate vaccinations for COVID-19 but only the offending vaccine and vaccines containing the component responsible for the allergy”. IgE-dependent allergy is due to the fine recognition of a chemical structure and/or a shape in space. “The allergy assessment makes it possible to identify patients at real risk of developing an anaphylactic reaction in the event of further exposure. The revaccination will then depend on this allergy assessment” [26].

Pfizer’s vaccine contains the active substance BNT162b2 (highly purified, 5’capped single-stranded messenger RNA, generated by in vitro transcription under cell-free conditions from the corresponding DNA encoding the spike protein of SARS-CoV-2), no adjuvant, 4 lipids (ALC-0315 or (4-hydroxybutyl) azanediyl bis (2-hexyldecanoate)), ALC-0159 2 [(polyethylene glycol)-2000]-N, N-ditetradecylacetamide, 1,2-distearoyl-sn-glycero-3-phosphocholine, cholesterol) and several electrolytes and other compounds (potassium chloride, monobasic potassium phosphate, sodium chloride, dibasic sodium phosphate dihydrate, sucrose). Clinical trials for registration of this vaccine have excluded people with a history of allergies to a vaccine. People who have had allergic reactions to foods or drugs in the past were not excluded, but may be under-represented. However, to date, nothing has confirmed the causes of the reactions of the two English cases [27] and 33 other American cases described since 3 January 2021 [28]. Thirty of these 35 people had a history of atopic disease, particularly asthma, but also other allergies or anaphylaxis. The CDC has proposed the specific procedures according to the risk stratification and the National Institute of Allergy and Infectious Diseases (NIAID) is establishing a three-year cohort. Caution is therefore required and surveillance should be strengthened in order to measure this risk, to understand the mechanisms involved, to put in place the necessary preventive measures (30 min of surveillance in the event of an anaphylaxis whatever the cause instead of the usual 15 min because most of anaphylactic reactions occur during this period, documentation of any reaction) and to accompany the scientific information that will be provided to the general public. The National Academy of Medicine brings together all the skills to participate in these actions. Although the Pfizer-BioNTech vaccine contains several excipients, PEG 2000 is the only one reported to cause anaphylaxis. All mRNA vaccines are likely to contain PEG which is used to stabilize the lipid nanoparticles [29].

## 3. Recommendations of the French Allergy Community and the Montpellier World Health Organization Collaborating Center

### 3.1. The Montpellier WHO Collaborating Center

In June 2018, the **WHO Collaborating Center (WHO CC) for the Scientific Classification of Allergic and Hypersensitivity Diseases** was established at the University Hospital of Montpellier, headed by Luciana Kase Tanno and Pascal Demoly [30]. This designation is the result of recognition by WHO of all the efforts of *the ALLERGY*
*in ICD-11 initiative* [10,11,12,31,32,33,34,35] and is intended to provide academic, research and scientific support to WHO in the implementation, refinement and maintenance of the WHO-FIC (Family of International Classifications) in the areas of our expertise. WHO CCs are institutions designated by the Director-General of the WHO and endorsed by the national minister of health to carry out activities in support of the WHO programs, such as communicable diseases, nutrition, mental health, and occupational health among others. Currently, there are 25 WHO CCs responsible for the WHO-FIC and the Montpellier WHO CC is the only one with expertise in allergy and clinical immunology.

### 3.2. Recommendations of the Montpellier World Health Organization Collaborating Center

Healthcare professionals must follow local authorities and policies related to indications and contra-indications for vaccines against COVID-19. Together with drug allergy specialists of the French allergy community under the aegis of the French Allergy Society (SFA), we provided recommendations regarding this important issue. All the cases of allergic reaction following COVID-19 vaccine should be recognized and treated accordingly (Figure 2). Serum (or plasma) levels of total tryptase and mature tryptase measurements are recommended in the diagnostic evaluation of anaphylaxis in order to support the diagnosis of anaphylaxis and exclude possible differential diagnosis, such as mast cell activation disorders. Table 1 provides an outline for precautions in the case of a personal history of allergies. The only recommendation should be addressed to patients who experienced prior allergic reaction to the vaccine in question or its components or those patients with prior allergy to an mRNA-based COVID-19 vaccine (Table 2). These cases should be referred to the allergists for risk stratification. Since there are intravenous drugs containing PEG, the French Allergy Society also recommends that all patients with personal history of anaphylaxis due to an unidentified intravenous drug should also be referred to the allergist for further investigation and specific recommendation (Figure 1).

There is currently limited data regarding the risk of anaphylaxis due to the COVID-19 vaccine in patients with mast cell activation syndrome or mastocytosis. Therefore, we do not contra-indicate the vaccination in this particular population [36,37,38]. Safety measures, such as premedication and post-vaccine observation should be individualized [37]. We also highlight that premedication with antihistamines and/or systemic steroids is insufficient to prevent immune-mediated anaphylactic reactions.

### 3.3. First Nation-Wide Post-Marketing Pharmacovigilant Data

Since the initial UK reports, in which two cases of anaphylaxis after mRNA Pfizer/BioNTech COVID-19 vaccine with only for one patient confirmed to be allergic to PEG [27], other reports have followed, without allergy work up so far.

The US CDC estimated that anaphylaxis to the mRNA COVID-19 vaccines would occur in 2.5 to 11.1 cases per million of doses, largely in individuals with a history of allergy [39]. Blumenthal et al. observed from all 64,900 subjects who received the first dose of mRNA COVID-19 vaccine, 16 developed confirmed anaphylaxis (0.025%, (95%CI, 0.014–0.040%)), seven cases from the Pfizer/BioNTech vaccine (0.02% (95%CI, 0.011%–0.056%)) and nine from Moderna vaccine (0.023% [95%CI, 0.011–0.044%]) [40].

The last report from the French ANSM reports a lower number of cases. From an overall 13,610,000 doses in 8 April 2021, 9,889,000 were Pfizer/BioNTech vaccines, 994,000 were Moderna vaccine and 2,725,089 AstraZeneca vaccine. Sixty-seven severe hypersensitivity reactions were reported (0.0005% per application), 58 cases (5.86 per million) from Pfizer/BioNTech vaccine, four cases (4.02 per million) from Moderna vaccine and five cases (1.83 per million) from the AstraZeneca vaccine [17].

Difficulties in recognition anaphylaxis, in particular of less severe cases, can hamper the collection of accurate epidemiological data of reported cases.

## 4. Let’s Keep Fighting COVID-19!

The COVID-19 vaccination is rolling-out vastly in different countries and surveillance programs are key to monitor severe adverse reactions, such anaphylaxis. Anaphylaxis due to vaccine is rare and specific cases should receive individualized investigation and care. The here presented recommendations are not from the WHO but have been offered to the WHO in order to support the vaccination program and intends to support to healthcare professionals in their daily basis.

## Figures and Tables

**Figure 1 vaccines-09-00560-f001:**
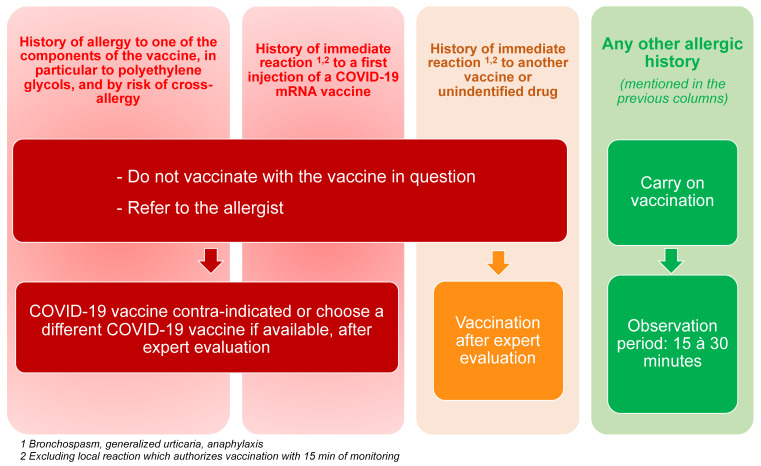
General recommendations proposed by the French Allergy Society (SFA) and French Federation of Allergy (FFAL) and endorsed by the Montpellier WHO Collaborating Center.

**Figure 2 vaccines-09-00560-f002:**
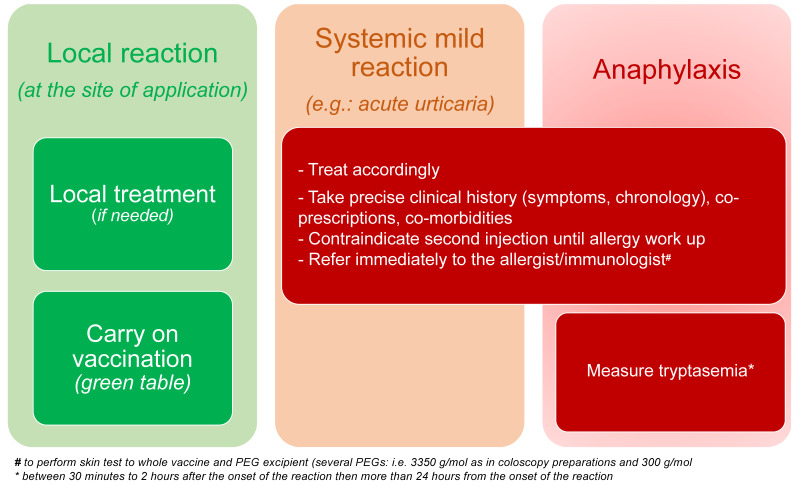
Management of acute reaction during the SARS-CoV-2 vaccination.

**Table 1 vaccines-09-00560-t001:** Procedures to the SARS-CoV-2 vaccination according to the personal history of allergies.

	Proceed with Vaccination	Special Precautions	Vaccination Contra-Indicated or Needing High Precaution
Patients’ Personal History	Prior history of allergic reaction to an identified food or venom, or defined group of medication not present in SARS-CoV-2 vaccine Inhalant allergy Family history of allergies Local (non-systemic) reaction to prior vaccination Contact dermatitis Chronic spontaneous urticaria	History of immediate allergic reactions to multiple drug classes, with the trigger unidentified ^1^ History of “immediate allergy” or anaphylaxis to a vaccine or a parenteral biological, an injected corticosteroid, colonoscopy preparation, or laxatives History of idiopathic anaphylaxis Possible mast cell activation syndrome (MCAS) / mastocytosis History of latex allergy or topical disinfectant allergy, bradykinin related angiedema or inhibitor angiotensin agent induced anaphylaxis	Prior allergic reaction to the vaccine in question For an mRNA-based COVID-19 vaccine, prior allergic reaction to another mRNA vaccine Prior allergic reaction to a component of the vaccine or drug including PEG (see Table 2)
Recommendations	Proceed with vaccination as normal, according to local guidelines An observation period of 15–30 min is advisable.	Specific allergy risk assessment (*) and possibility of PEG allergy (**) Consider referral to allergist-immunologist Consider observation for 30 min if vaccination proceeds and at the allergy department	Do not vaccinate with the vaccine in question Ideally choose a different COVID-19 vaccine if available and not contra-indicated Consider referral to allergist-immunologist for specific allergy risk assessment (*) Consider shared decision making for future vaccination or use of drugs containing proven allergy component

*^1^-This may indicate PEG = polyethylene glycol (PEG) allergy.* Specific allergy risk assessment before vaccination (*): Allergy work-up should be considered individually according to the clinical history of each patient. Specific allergy risk assessment for PEG allergy (**): several PEGs: i.e., 3350 g/mol as in coloscopy preparations and 300 g/mol) through skin test: NEGATIVE skin test: proceed *SARS-CoV-2* vaccination as normal. POSITIVE skin test: mRNA-based COVID-19 vaccine should be contra-indicated, consider use of other *SARS-CoV-2 vaccine without PEG (when available)* at the allergy department, *decision for desensitization to drugs containing PEG should be taken individually.*

**Table 2 vaccines-09-00560-t002:** Current (January 2021) available SARS-CoV-2 Vaccines and their excipients.

Vaccine & Manufacturer	Vaccine Type	Excipients
CoronaVac (Sinovac, China)	Inactivated vaccine (formalin with aluminum adjuvant)	Aluminum hydroxide, disodium hydrogen phosphate, sodium dihydrogen phosphate, sodium chloride
Convidicea Ad5-nCoV (CanSino Biologics, Beijing Inst. Biotech., NPO Petrovax)	Recombinant adenovirus type 5 vector against spike RBD protein	N/A
BBIBP-CorV (Sinopharm, Beijing Institute & Wuhan Inst. of Biological Products)	Inactivated SARS-CoV-2 (vero cells) + aluminum hydroxide adjuvant	Aluminum hydroxide, disodium hydrogen phosphate, sodium dihydrogen phosphate, sodium chloride, sodium hydroxide, sodium bicarbonate, M199
Pfizer-BioNTech BNT162b2	mRNA-based vaccine (encoding the viral spike (S) glycoprotein)	(4-hydroxybutyl) azanediyl)bis (hexane-6,1-diyl)bis(2-hexyldecanoate)] (ALC-0315), 2-[(**polyethylene glycol**)-2000]-N,N-ditetradecylacetamide (ALC-0159), 1,2-Distearoyl-sn-glycero-3-phosphocholine cholesterol, potassium chloride, potassium dihydrogen phosphate, sodium chloride, disodium hydrogen phosphate dihydrate, sucrose, water for injection
Moderna mRNA-1273	mRNA-based vaccine (encoding the pre-fusion stabilized spike (S) glycoprotein)	Lipids (SM-102, 1,2-dimyristoyl-rac-glycero3-methoxy**polyethylene glycol**-2000 [PEG2000-DMG], cholesterol, and 1,2-distearoyl-snglycero-3-phosphocholine [DSPC]), tromethamine, tromethamine hydrochloride, acetic acid, sodium acetate, and sucrose.
ChAdOx1 (Oxford/AstraZeneca; Covishield in India)	Replication-deficient viral vector vaccine (adenovirus from chimpanzees)	L-Histidine, L-Histidine hydrochloride monohydrate, Magnesium chloride hexahydrate, **polysorbate 80**, Ethanol, Sucrose, Sodium chloride, Disodium edetate dihydrate, Water for injection
Covaxin (BBV152) (Bharat Biotech, India)	Inactivated vaccine	N/A
Sputnik V (Gamaleya Research Inst)	Non-replicating, two-component vector (adenovirus) against spike (S) glycoprotein	Tris (hydroxymethyl) aminomethane, sodium chloride, sucrose, magnesium chloride hexahydrate, Sodium EDTA, **polysorbate 80**, ethanol, water for injection
EpiVacCorona (Federal Budgetary Research Institution State Research Ctr, Russia)	Peptide vaccine with aluminum adjuvant	Aluminum hydroxide, potassium dihydrogen phosphate, potassium chloride, sodium hydrogen phosphate dodecahydrate, sodium chloride, water for injection
Ad26.COV2.S (Jansen)	Adenovirus 26 vectored vaccine using AdVac and PER.C6 technology	Sodium chloride, citric acid monohydrate, **polysorbate 80**, 2 hydroxypropyl-B-cyclodextrin (HBCD), ethanol (absolute), sodium hydroxide
SARS-CoV-2 vaccine formulation with adjuvant (Sanofi Pasteur and GSK)	SARS-CoV-2 vaccine formulation with adjuvant (S-protein) (Baculovirus production) Spike protein	Sodium phosphate monobasic monohydrate, sodium phosphate dibasic, sodium chloride **polysorbate 20**, disodium hydrogen phosphate, potassium dihydrogen phosphate, potassium chloride

## Data Availability

Review document based on published data.

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
