# Peer review of "SARS-CoV-2 Vaccination and Anaphylaxis: Recommendations of the French Allergy Community and the Montpellier World Health Organization Collaborating Center"

_vaccines, 2021, doi:10.3390/vaccines9060560_

Round 1

Reviewer 1 Report

general
anaphylaxis is only one serious adverse event after corona vaccination. one othe is the sinus vein thrombosis. please give a judgement what is more of concern, anaphylaxis or sinus vein thrombosis ... . 
suggestions
1. the paper is too long - make it short for exampüle, the paragraph line 273-279 could be cancelled in total. 
special
line 24 and line 332: you mean "courtiers" or countries ... ?
line 100: "thanks" to yourself sounds somewhat strange ... 
line 137; you mention "few" reports ... how manny ... exactly ? 
line 167: what is "DPT" .... ?
line 169: what is "MMR" ... ? 
line 190: you probably want to say "cross-react" not only "cross" 
line 251: give a reason whay 30 minutes and not 4 hours ... 
line 281: you probably mean "local authorities" not "authorizations"
figure 2: explain why "tryptasemia" should be measured 
table 1: make it more clear that the upper part is HISTORY and the lower part is PROCEDERE
table 2: "alum" means probalby aluminium ... htere is enough space
line 342 and 343: what is APHITEM ... CHUM ... AllerGOS ... are these companies ... 

fr 14 may 2021 

Author Response

Point-by-point response to reviewers

Date: 25.05.21

Manuscript Number: vaccines-1234557

Title of Article: SARS-CoV-2 Vaccination & Anaphylaxis: Recommendations of the French Allergy Community & the Montpellier World Health Organization Collaborating Centre

We would like to thank very much the editor and reviewers for their time, and comments, which have resulted in substantial improvement of our work.

REVIEWER 1

Comments and Suggestions for Authors

general
anaphylaxis is only one serious adverse event after corona vaccination. one othe is the sinus vein thrombosis. please give a judgement what is more of concern, anaphylaxis or sinus vein thrombosis ... . 

The reviewer is correct. There are no doubts regarding the severity of sinus vein thrombosis, but decoded to cover anaphylaxis due to the expertise of the authors in the field and in response to many questions raised by patients and health professionals

suggestions
1. the paper is too long - make it short for exampüle, the paragraph line 273-279 could be cancelled in total. 
special
line 24 and line 332: you mean "courtiers" or countries ... ?
line 100: "thanks" to yourself sounds somewhat strange ... 
line 137; you mention "few" reports ... how manny ... exactly ? 
line 167: what is "DPT" .... ?
line 169: what is "MMR" ... ? 
line 190: you probably want to say "cross-react" not only "cross" 
line 251: give a reason whay 30 minutes and not 4 hours ... 
line 281: you probably mean "local authorities" not "authorizations"
figure 2: explain why "tryptasemia" should be measured 
table 1: make it more clear that the upper part is HISTORY and the lower part is PROCEDERE
table 2: "alum" means probalby aluminium ... htere is enough space
line 342 and 343: what is APHITEM ... CHUM ... AllerGOS ... are these companies ... 
Many thanks for the suggestions. The text has been tuned accordingly.

Reviewer 2 Report

General comments

This paper is a review of SARS-CoV-2 vaccination and what is known about anaphylaxis caused by such vaccines at present.

The paper is written by the team at the Montpellier allergy and asthma center, a WHO collaborating center. This gives the paper the character of a WHO position paper, although not clearly stated. It should be clearly stated if the paper has been or will be accepted as the standpoint of WHO.

MPG has been used for the formulation of several drugs, e.g. interferons, without any major general reactions reported. In the 1980-ties, Pharmacia Diagnostics developed a series of mPEG modified allergen extracts. The products were withdrawn due to the development of positive skin reactions as reported by Freddie Hargreaves group at McMasters (1) and Mosbech in Copenhagen (2-5) and the non-finalized ongoing trials summarized by Dreborg and Åkerbom (6). No severe reactions occurred. However, IgG anti-PEG IgG as well as in some cases even IgE antibodies developed.

The name of the vaccines should be the same throughout the manuscript and the abbreviation stated after the first mention. It is sometimes not clear if both or only one of the mRNA vaccines is alluded to. As it is now, it is not clear in many instances. It would be of value to have a separate abbreviation when both the mRNA vaccines are alluded at.

There are no instructions for the investigation of possible anaphylaxis reactions caused by these vaccines. To gather similar data form many different sources, I think it would be of value to give some directions for skin prick and intradermal tests e.g. establishing the highest concentration of the respective compound not causing local reactions in vaccine naive volunteers, or refer to presently available instructions.

Specific comments

Line            Comment

43-4            I propose: for the Pfizer-BioNTech mRNA COVID-19 vaccine (Pfizer vaccine) and similarly the Moderna mRNA COVID-19 vaccine (Moderna vaccine). To be listed as abbreviations. In addition I propose that mRNA COVID-19 vaccineS (mRNA vaccineS) is used when both are alluded at. Have a look at all your abbreviations!

44-5            “ … regulatory agencies, such as the European Commission, … “. Isn’t EMA the regulatory body, the commission an administrative authority?

47               I propose: “ … in USA. Until April 29021, 4,493 … “.

 59              Later – than what? Another proposal: “On December 30th 2020, the UK …  “.

68               “inhalant allergens”(?).

191             A systematic review? Reference!

216             A little late – RNA (mRNA) has been used previously

226             Delete “possibly”.

342             Lusiana Kase Tanno is is abbreviated LK as well as LKT. Go through all your abbreviations, even APHITEM and CHUM.

Table 1.      The table must be reorganized. Maybe the manuscript was correct but the final table is not acceptable. Using electric illumination (Kelvin?) I had difficulties to read the white text on yellow background. I think the text can be better arranged by using landscape page format.

Table 2       The table must be reorganized. There are some empty rows. It is difficult to read the text about the last vaccine.

References

  1. Juniper EF, Roberts RS, Kennedy LK, O'Connor J, Syty-Golda M, Dolovich J, et al. Polyethylene glycol-modified ragweed pollen extract in rhinoconjunctivitis. J Allergy Clin Immunol. 1985;75(5):578-85.
  2. Mosbech H, Dirksen A, Dreborg S, Frolund L, Heinig JH, Svendsen UG, et al. Hyposensitization in asthmatics with mPEG-modified and unmodified house dust mite extract. IV. Occurrence and prediction of side effects. Allergy. 1990;45(2):142-50.
  3. Dreborg S, Akerblom EB. Immunotherapy with monomethoxypolyethylene glycol modified allergens. Critical reviews in therapeutic drug carrier systems. 1990;6(4):315-65.

Author Response

Point-by-point response to reviewers

Date: 25.05.21

Manuscript Number: vaccines-1234557

Title of Article: SARS-CoV-2 Vaccination & Anaphylaxis: Recommendations of the French Allergy Community & the Montpellier World Health Organization Collaborating Centre

We would like to thank very much the editor and reviewers for their time, and comments, which have resulted in substantial improvement of our work.

REVIEWER 2

Comments and Suggestions for Authors

General comments

This paper is a review of SARS-CoV-2 vaccination and what is known about anaphylaxis caused by such vaccines at present.

The paper is written by the team at the Montpellier allergy and asthma center, a WHO collaborating center. This gives the paper the character of a WHO position paper, although not clearly stated. It should be clearly stated if the paper has been or will be accepted as the standpoint of WHO.

Many thanks for the suggestions. A statement has been included accordingly (345-348).

MPG has been used for the formulation of several drugs, e.g. interferons, without any major general reactions reported. In the 1980-ties, Pharmacia Diagnostics developed a series of mPEG modified allergen extracts. The products were withdrawn due to the development of positive skin reactions as reported by Freddie Hargreaves group at McMasters (1) and Mosbech in Copenhagen (2-5) and the non-finalized ongoing trials summarized by Dreborg and Åkerbom (6). No severe reactions occurred. However, IgG anti-PEG IgG as well as in some cases even IgE antibodies developed.

The text has been added accordingly (217-223).

The name of the vaccines should be the same throughout the manuscript and the abbreviation stated after the first mention. It is sometimes not clear if both or only one of the mRNA vaccines is alluded to. As it is now, it is not clear in many instances. It would be of value to have a separate abbreviation when both the mRNA vaccines are alluded at.

The reviewer is correct. Vaccines’ names have been standardized across the text.

There are no instructions for the investigation of possible anaphylaxis reactions caused by these vaccines. To gather similar data form many different sources, I think it would be of value to give some directions for skin prick and intradermal tests e.g. establishing the highest concentration of the respective compound not causing local reactions in vaccine naive volunteers, or refer to presently available instructions.

Many thanks for the suggestions. The proposed document is addressed to a broader audience than allergists reason why there was a decision to avoid details in terms of concentrations of skin tests.

Specific comments

Line            Comment

43-4            I propose: for the Pfizer-BioNTech mRNA COVID-19 vaccine (Pfizer vaccine) and similarly the Moderna mRNA COVID-19 vaccine (Moderna vaccine). To be listed as abbreviations. In addition I propose that mRNA COVID-19 vaccineS (mRNA vaccineS) is used when both are alluded at. Have a look at all your abbreviations!

Many thanks for the suggestion. The text has been tuned accordingly.

44-5            “ … regulatory agencies, such as the European Commission, … “. Isn’t EMA the regulatory body, the commission an administrative authority?

47               I propose: “ … in USA. Until April 29021, 4,493 … “.

 59              Later – than what? Another proposal: “On December 30th 2020, the UK …  “.

68               “inhalant allergens”(?).

191             A systematic review? Reference!

216             A little late – RNA (mRNA) has been used previously

226             Delete “possibly”.

342             Lusiana Kase Tanno is is abbreviated LK as well as LKT. Go through all your abbreviations, even APHITEM and CHUM.

Many thanks for the suggestions. The text has been tuned accordingly.

Table 1.      The table must be reorganized. Maybe the manuscript was correct but the final table is not acceptable. Using electric illumination (Kelvin?) I had difficulties to read the white text on yellow background. I think the text can be better arranged by using landscape page format.

Table 2       The table must be reorganized. There are some empty rows. It is difficult to read the text about the last vaccine.

Tables have been reformatted in order to make the information clearer.

References

  1. Juniper EF, Roberts RS, Kennedy LK, O'Connor J, Syty-Golda M, Dolovich J, et al. Polyethylene glycol-modified ragweed pollen extract in rhinoconjunctivitis. J Allergy Clin Immunol. 1985;75(5):578-85.
  2. Mosbech H, Dirksen A, Dreborg S, Frolund L, Heinig JH, Svendsen UG, et al. Hyposensitization in asthmatics with mPEG-modified and unmodified house dust mite extract. IV. Occurrence and prediction of side effects. Allergy. 1990;45(2):142-50.
  3. Dreborg S, Akerblom EB. Immunotherapy with monomethoxypolyethylene glycol modified allergens. Critical reviews in therapeutic drug carrier systems. 1990;6(4):315-65.

The references are now included.
